# Impacts of Sodium Arsenite on Wood Microbiota of Esca-Diseased Grapevines

**DOI:** 10.3390/jof7070498

**Published:** 2021-06-22

**Authors:** Emilie Bruez, Philippe Larignon, Christophe Bertsch, Guillaume Robert-Siegwald, Marc-Henri Lebrun, Patrice Rey, Florence Fontaine

**Affiliations:** 1UR Œnologie, Université de Bordeaux, ISVV, 33882 Villenave d’Ornon, France; 2IFV, Pôle Rhône-Méditerranée, 30230 Rodilhan, France; philippe.larignon@vignevin.com; 3Laboratoire LVBE, Université de Haute-Alsace, 68000 Colmar, France; christophe.bertsch@uha.fr; 4INRAE, Université Paris-Saclay, AgroParisTech, UMR BIOGER, 78850 Thiverval-Grignon, France; guillaume.robert-siegwald@inrae.fr (G.R.-S.); marc-henri.lebrun@inrae.fr (M.-H.L.); 5Bordeaux Sciences Agro, INRAE, Université de Bordeaux, UMR SAVE, 33170 Villenave d’Ornon, France; patrice.rey@agro-bordeaux.fr; 6RIBP EA 4707-USC INRAE 1488, SFR Condorcet FR CNRS 3417, Université Reims Champagne-Ardenne, 51687 Reims, France; florence.fontaine@univ-reims.fr

**Keywords:** fungicide, trunk disease, white rot, *Fomitiporia mediterranea*, metabarcoding

## Abstract

Although sodium arsenite was widely used in Europe until its ban in 2003, its effects on microorganisms is not clearly understood. To improve our understanding of sodium arsenite curative effect on GTDs, grapevines displaying esca-foliar symptoms from different French regions (Alsace, Champagne, Languedoc) were treated or not with sodium arsenite, and analyzed for their wood microbiota. Using metabarcoding, we identified the fungal and bacterial taxa composition of microbiota colonizing woody trunk tissues. Large differences in fungal microbiota composition between treated and untreated grapevines were observed while no major impacts were observed on bacteria microbiota. The main fungal species detected in untreated necrotic woody tissues was *Fomitiporia mediterranea* (63–94%), a fungal pathogen associated with esca. The relative abundance of this fungal species significantly decreased after sodium arsenite treatment in the three vineyards, in particular in white-rot necrotic tissues and their borders (−90%). *F. mediterranea* was the most sensitive to sodium arsenite among fungi from grapevine woody tissues. These results strongly suggest that the effect of sodium arsenite on GTDs is due to its ability to efficiently and almost specifically eliminate *F. mediterranea* from white-rot necrotic tissues, allowing saprobic fungi to colonize the tissues previously occupied by this pathogenic fungus.

## 1. Introduction

Although various fungicides or biocontrol agents control the grapevine foliar diseases (downy mildew, *Plasmopara parasitica*; powdery mildew, *Erysiphe necator*; gray mold, *Botrytis cinerea*), there is currently no efficient method against Grapevine Trunk Diseases (GTDs), a threat for viticulture worldwide ever since the ban of sodium arsenite in the 2000s [1,2,3]. Esca, Botryosphaeria and Eutypa diebacks, in particular, are the most frequent and devastating GTDs in Europe [4]. In Italy, the incidence of GTDs ranges from 8% to 19% on 15–18-year-old grapevines, while peaking at 10% in Spain. In France, 13% of grapevines have become unproductive, inducing yield losses estimated at some EUR 1 billion per year [5]. GTDs are caused by a broad range of taxonomically unrelated fungal pathogens, including *Phaeomoniella chlamydospora*, *Phaeoacremonium minimum*, *Eutypa lata*, *Fomitiporia mediterranea*, and *Botryosphaeriaceae* species (*Diplodia seriata*, *Diplodia mutila*, and *Neofusicoccum parvum*) [3,6,7,8]. These fungi colonize grapevine trunk and cordon woody tissues, causing various types of necrosis, which alters grapevine phenol metabolism [9]. *P. chlamydospora*, for instance, is frequently isolated from black punctuations and sectorial necrotic tissues, whereas *F. mediterranea* is mainly isolated from sectorial and central white-rot necrotic tissues [7,10,11].

Arsenic compounds were used in the 18th century as insecticides to protect plants against pests [12,13]. In viticulture, those compounds were mostly used to control insects and the fungi involved in anthracnose, as well as the fungi, *Phomopsis* spp., responsible for cane and leaf spot diseases [14,15,16,17,18]. Sodium arsenite was subsequently used, from the beginning of the 20th century, to control esca [19,20,21,22,23,24]. Although sodium arsenite proved very effective in controlling esca, its mode of action is still poorly understood. Larignon et al., using in vitro and microbiological methods, observed that the treatment of grapevines with sodium arsenite strongly decreased the occurrence of plant pathogenic fungi such as *P. chlamydospora* and *F. mediterranea* in trunks from esca-diseased plants [25]. In addition, sodium arsenite displayed a strong fungicide activity on several plant pathogenic fungi, such as *F. mediterranea* and *Botryosphaeria* spp., isolated from esca-diseased grapevine trunks [25]. These observations suggest that sodium arsenite acts as a curative nonselective fungicide on trunk fungi [23]. Our project aimed at developing insights into sodium arsenite’s effects on esca by first studying its effects on grapevine physiology [26] and subsequently on woody microbiota (the present study). The long-term goal of this project is to identify products that would have the same effect on GTD fungi as sodium arsenite by mimicking its mode of action.

To improve our understanding of sodium arsenite curative effect on grapevine trunk diseases, we analyzed its impact on fungal and bacterial wood microbiota of grapevines displaying esca-foliar symptoms from three French vineyards (Alsace, Champagne and Languedoc). We also assayed the sensitivity of different fungal species isolates from grapevine wood tissues to sodium arsenite. Overall, this study highlights the strong impact of sodium arsenite on the fungal community colonizing woody tissues of grapevine trunks with esca-disease symptoms

## 2. Materials and Methods

### 2.1. Plant Material and Sampling

The sampling sites were located in three French vineyards, each cultivated with a specific cultivar ‘Gewürztraminer’ with rootstock 33EM, in Alsace (Rouffach, GPS coord.: 47°57′56″ N, 07°17′13″ E); ‘Chardonnay’ with rootstock SO4 or Fercal, in Champagne (Avize, GPS coord.:48°58′29″ N, 04°00′46″ E), and ‘Merlot’ with rootstock SO4, in Languedoc (Rodilhan, GPS coord: 43°82′99″ N, 04°45′02″ E). These established vineyards, aged 25 for Alsace, 27 for Champagne, and 40 for Languedoc, were examined for esca-foliar symptoms in both 2013 and 2014. In 2013 and 2014, before sodium arsenite treatment in 2014 and 2015, 10 plants expressing esca-foliar symptoms were selected at random in each vineyard. Five of the 10 selected plants were treated with sodium arsenite (Pyralesca RS) at 1250 g/hL until streaming, at the end of winter after pruning, and before bud bursting, BBCH 00 [27]. The treated and control grapevines were uprooted in September 2014 and 2015, respectively. Their trunks were cut transversally and the status of the wood, necrotic or healthy, recorded. Metabarcoding analyses were performed on white-rot necrotic tissues of grapevines from Alsace and Champagne, on border tissues from all three regions, and on the sectorial black necrotic tissues from Languedoc (Table 1). Non-necrotic tissue of grapevines from all three regions was analyzed (Table 1). Although white-rot necrotic tissues were not present in sufficient amounts in Languedoc grapevine to allow the DNA to be extracted, their sectorial black necrotic tissues were analyzed. In all, 36 DNA samples were obtained (Appendix A).

### 2.2. DNA Extraction and Analysis of Fungal and Bacterial Communities

The DNA extraction protocol used here was similar to that of Bruez et al. [11,28]. SSCP analysis was performed on samples collected both in September 2014 and September 2015. The pair of primers recognizing the mitochondrial large subunit rDNA gene, ML1-ML2 from White et al. was selected for fungal SSCP [28]. The PCR protocol was similar to that of Bruez et al. [10]. A pair of primers recognizing the V5–V6 region of the 16S rRNA gene was used for bacterial SSCP (799f/1115r) in accordance with Bruez et al. [28]. The metabarcoding protocol used was similar to that of Bruez et al. [26]. The composition of grapevine wood microbial communities was characterized by tag-encoded 454 pyrosequencing (fungi: ITS; bacteria: 16S). For fungi, the pair of primers, ITS1 and ITS4F [29], was used to amplify the ITS1 and ITS2 regions of the nuclear ribosomal repeat unit. For bacteria, PCR amplifications targeted the V5-V6 variable region of the 16S rDNA gene. 799f and 1115r was chosen, as that pair of primers does not amplify chloroplast DNA [30].

### 2.3. Post-Run Analysis

Fungal and bacterial data were processed using the Quantitative Insights Into Microbial Ecology (QIIME) pipeline (version 1.7.0) [31] (https://submit.ncbi.nlm.nih.gov/subs/sra/SUB8307285/metadata) (accessed on 1 April 2021). In brief, sequences were selected in accordance with the following initial criteria: (i) ≥450 nucleotides in length; (ii) a perfect match to the primers and barcodes; with (iii) no ambiguous base allowed. Although the bacterial primers were designed to amplify bacterial DNA, we verified, using Metaxa software, that no sequences corresponding to chloroplast or mitochondria DNAs were present in the bacterial dataset [32]. The V-Xtractor and FungalITSextractor tools were used, respectively, to extract the V5-V6 region of the bacterial 16S rDNA gene and the fungal ITS1 region [33]. Microbial sequences were binned into OTUs, using a 97% identity threshold with the UPARSE algorithm implemented in the USEARCH pipeline, with the most abundant sequence from each OTU being selected as a representative sequence for that OTU (Appendix A) [34]. Taxonomy was assigned to OTUs, using the Basic Local Alignment Search Tool (BLASTN) for each representative sequence against the GreenGenes reference database for bacteria (97% similarity), and the Fungal ITS Database produced by Nilsson and co-workers (99% similarity) [35,36]. The species/genus assignment of the 20 most abundant fungal OTUs were confirmed using BLASTN at NCBI platform. The OTU was assigned to a species/genus from NCBI database when the percentage identity was higher than 99% and the coverage higher than 895% (Appendix A). Additional BLASTN searches were performed at Mycobank database (https://www.mycobank.org) (accessed on 1 April 2021) and UNITE database [35,36] to confirm the species/genus assignment defined using BLASTN at NCBI. Phylogenetic trees provided by NCBI and Mycobank were used to test if the assigned species was the only species clustering with the analyzed OTU (Appendix A). Sequences were deposited under SRA number PRJNA732838 (https://www.ncbi.nlm.nih.gov/sra/PRJNA732838) (accessed on 1 June 2021).

### 2.4. Statistical Analysis

All the statistical analyses were carried out using the R statistical software, version 3.1.1. OTU-distribution matrixes were used to perform rarefaction analyses and to calculate diversity (Shannon and Simpson) indexes with the EstimateS software package [37]. To compare different indexes between samples, an analysis of variance and a Kruskal–Wallis test were performed. Using the Vegan package, a non-metric multidimensional scaling (nMDS) was performed based on Bray–Curtis dissimilarity, and validated with an ANOSIM test.

### 2.5. In Vitro Fungal Growth Inhibition Assays

Sodium arsenite (Pyralesca RS) was suspended in sterile water and added to molten malt-agar medium poured into Petri dishes (agar: 20 g/L, malt: 15 g/L; MA) at a concentration ranging from 1 to 500 mg/L, with MA as control. Each plate was inoculated with a 6 mm diameter mycelial plug from the actively growing margins of each tested fungal strain grown on MA. The culture was left at room temperature in the dark for 7–14 days, depending on the growth rate of the fungus tested. When the margin of the colony growing on MA without sodium arsenite (control) approximately was at 1 or 2 cm from the edge of the Petri dish, mycelial growth was recorded by measuring colony diameter. The growth tests in this study were performed in 2014 on fungi isolated from the trunks of treated and untreated grapevines (Appendix A). The fungi were either (i) pathogenic ones associated with GTDs, such as *Phaeomoniella chlamydospora*, *Fomitiporia mediterranea*, and *Eutypa lata* or (ii) saprobes such as *Trichoderma* spp. and *Penicillium* spp. [7,38,39]. All these isolates were assayed for their sensitivity to sodium arsenite. Mycelial growth rate was assessed on malt agar plates, amended or not with increasing concentrations of sodium arsenite. Growth inhibition was estimated using the following formula: 100 × (diameter of the control colony—diameter of the colony on fungicidal medium)/diameter of the control colony). The concentration required to obtain a 50% (EC50) or 90% (EC90) inhibition of mycelial growth was estimated, using a linear regression analysis in which X = the log of fungicide concentration, and Y = the probit % of the control. The values of the lethal concentration were defined in accordance with the following methodology: those mycelial plugs not grown with fungicide on the culture medium were transferred onto an MA culture medium without fungicide. The mycelial growth originating from the plug was recorded (no growth = lethal).

## 3. Results

### 3.1. Sampling Grapevine Trunk Woody Tissues

The vineyards studied were located in Alsace, Champagne, and Languedoc. The grapevine cultivars sampled in each region were ’Gewürztraminer’, ’Chardonnay’, and ’Merlot’, respectively. In 2014 and 2015, 10 plants expressing esca-foliar symptoms were selected at random in each vineyard. Half of the plants were treated with sodium arsenite at the end of winter. All 10 plants, treated or not, were uprooted in September 2014 and 2015, respectively. All untreated plants displayed both esca-foliar symptoms, and central white-rot in woody tissues (WR, Table 1). WR tissues were observed in all plants from Alsace (100%), in 80% of those from Champagne, and in 60% of those from Languedoc (Table 1). The woody tissues of trunks from treated grapevines were similar (presence of white-rot necrosis, sectorial black necrosis, and border tissue) to those of untreated grapevines (Table 1). None of the treated plants, however, displayed esca-foliar symptoms in both experiments, whereas all the control plants expressed esca-foliar symptoms (Table 1). These results strongly suggest that the sodium arsenite treatment used in these assays was efficient to cure these old grapevines from esca. Metabarcoding analyses were performed on central white-rot necrotic tissues exclusively for grapevines from Alsace and Champagne; on tissues located between white-rot and non-necrotic tissues (borders) for grapevines from all three regions; on sectorial black necrotic tissues for Languedoc grapevines (Table 1). Non-necrotic tissues were analyzed for grapevines from all regions (Table 1).

### 3.2. Metabarcoding Analysis of Microbial Taxa from Esca-Diseased Grapevines

Thirty-six DNA samples (Appendix A) in all were used for amplicon-based metabarcoding of fungi (ITS1-ITS2) and bacteria (16S). The authors did not obtain border tissue results from Languedoc in 2014. In total 718 fungal OTUs and 3386 bacterial OTUs were identified by metabarcoding (>10 sequences) in woody tissues from esca-diseased grapevines sampled in the three vineyards in 2014 and 2015 (all samples, Appendix A). Fungal microbiota Shannon index [40] values ranged from 0.43 to 3.19 (Table 2), whereas Simpson index [39] values were lower than 0.97 (Appendix A). Although fungal communities were diversified, the number of abundant OTUs was low, with the 20 most abundant fungal OTUs (>1%), representing 99% of all sequences, and eight fungal OTUs constituting 80% of all sequences. Shannon index values for bacterial communities ranged from 0.37 to 5.40 (Table 2), whereas Simpson index values were lower than 0.98 (Appendix A). No significant differences were observed for bacterial and fungal communities among samples in 2014 and 2015 for Shannon and Simpson indexes. No significant differences were observed in Shannon and Simpson indexes for treated and untreated grapevines. Significant differences, however, were observed first for Simpson bacterial indexes: white-rot from Alsace and Champagne and sectorial necrosis from Languedoc (*p* = 0.0492; Appendix A); secondly, for Shannon bacterial and fungal indexes: white-rot (bacterial *p* = 0.007; fungal *p* = 0.049, Table 2), sectorial necrosis (bacterial *p* = 0.049; fungal index *p* = 0.049, Table 2). NMDS and ANOSIM analyses were performed to compare variations in taxa composition of microbiota isolated from the woody tissues of all grapevines, according to sampling year, type of woody tissue, region, and sodium arsenite treatment. As no significant differences were observed in taxa composition between grapevines sampled in 2014 and 2015 (fungal communities *p* = 0.502; bacterial communities *p* = 0.328; Appendix A), only samples from 2014 were further analyzed. The fungal microbiota was essentially the same in the different regions (Appendix A, *p* = 0.062). A comparison of the fungal microbiota between treated and untreated grapevines indicated major differences (Appendix A, *p* = 0.001). When focus was placed on the different types of tissue (non-necrotic, central white-rot necrosis, borders of white-rot necrosis, and sectorial black necrosis), further differences became apparent (Appendix A, *p* = 0.001). Bacterial communities showed differences for all three regions (Appendix A, *p* = 0.029). These differences did not depend on the simple distinction between treated and untreated grapevine (Appendix A, *p* = 0.203), but on the types of tissue involved (Appendix A, *p* = 0.001).

### 3.3. Metabarcoding Analysis of Fungal Taxa from Untreated Esca-Diseased Grapevines

As previously stated, fungal microbiota did not differ according to the region (*p* = 0.492; Appendix A), but there were significant differences between microbiota from different woody tissues: non-necrotic, white-rot necrosis, border necrosis, and sectorial necrosis (*p* = 0.001; Appendix A). SSCP fingerprint data also revealed differences (unshown data; *p* = 0.001) in fungal communities when different types of woody tissue from untreated grapevines were studied. The SSCP fingerprint data of microbiota from white-rot necrosis and sectorial necrosis tissues strongly differed from those of non-necrotic tissues according to the PCA axis 2 (Figure 1). Those from borders were intermediate between the fingerprint data of necrotic and non-necrotic tissues according to the PCA axis 2 (Figure 1). These results suggested that necrotic and non-necrotic fungal tissue communities differed, irrespective of the region.

The species/genus assignment of the 20 most abundant fungal OTUs was performed using a combination of BLASTN searches (NCBI, UNITE, Mycobank, Bethesda MD, USA) and phylogenetic analyses (see material and methods). Most OTUs could be assigned to a single fungal species without ambiguity including all GTD associated fungi detected in these samples (Appendix A). Five OTUs (OTU-1, OTU-2, OTU-5, OTU-14, OTU-18) were only assigned at the genus level, since their ITS1 sequences were not discriminant enough to identify the correct species, although three OTUs (OTU-1, OTU-2, OTU-14) could be assigned to a phylogenetic clade corresponding to a few related species (Appendix A). One OTU (OTU-4) corresponded to a still uncharacterized genus from the Agaricales identified in other environmental samples and related to the *Chontrostereum* genus.

In untreated non-necrotic woody tissues, *F. mediterranea* was the most abundant fungal species in Alsace and Champagne grapevines (32% and 77%, respectively, Table 3; Figure 2), and the second most abundant fungus in Languedoc grapevines (14%, Table 3; Figure 2). *Seimatosporium vitis* was the most abundant fungal species in non-necrotic woody tissues of untreated Languedoc grapevines (38%, Table 3; Figure 2), and the second most abundant species in non-necrotic woody tissues of untreated Alsace grapevines (23%, Table 3; Figure 2), whereas it was mostly absent from trunks of Champagne grapevines (0.3%, Table 3). The differences in the relative abundance of *S. vitis* across regions were significant (*p* = 0.035, Appendix A). Fungi found at lower relative abundance (4–10%, Table 3) in non-necrotic woody tissues from untreated grapevines included plant pathogens such as *Phaeomoniella chlamydospora* (8.8% Languedoc, 4.2% Alsace, 1.0% Champagne, Table 3), *Phaeoacremonium viticola* (4.1% Languedoc, 1.0% Alsace, 0.1% Champagne, Table 3), *Diplodia seriata* (7.8% Languedoc, 5.2% Alsace, 0.2% Champagne, Table 3), and *Eutypa lata* (0.2% Languedoc, 4.4% Alsace, 0.1% Champagne, Table 3). Saprobes were also detected at relatively low abundance (4.0–14%, Table 3) in non-necrotic woody tissues from untreated grapevines. These saprobes included *Aureobasidium pullulans* (0.1-4.0%), an unknown *Agaricales* genus (0.3–14%), *Cladosporium* spp. (0.1–11.4%), and *Mucor circinelloides* (0–9.8%). White-rot necrotic woody tissues of untreated Alsace and Champagne grapevines were almost exclusively colonized by *F. mediterranea* (92 and 94%, respectively, Table 3; Figure 2). *F. mediterranea* was also the most abundant fungal species in borders between white-rot necrotic and non-necrotic tissues of untreated grapevines from all regions (63% in Alsace; 81% in Champagne; 74% in Languedoc). In all these tissues, *F. mediterranea* was associated with *P. chlamydospora* as the second most abundant species (27% in Alsace; 16% in Champagne; 17% in Languedoc). These two fungi dominated the necrotic tissues (total of 90–95%, Table 3). Significant amounts of sectorial black necrotic tissues were observed in trunks of untreated Languedoc grapevines. Only three fungal species, *S. vitis*, *D. seriata*, and *F. mediterranea* colonized these necrotic tissues (56%, 18%, and 17%, respectively, for a total of 90%, Table 3).

### 3.4. Metabarcoding Analysis of Bacterial Taxa from Untreated Esca-Diseased Grapevines

Bacterial microbiota did not differ according to the region (Appendix A, *p* = 0.212) or tissue (Appendix A, *p* = 0.218). An additional statistical analysis of each species highlighted significant differences among tissues in the relative abundances of *Bradyrhizobium* spp., *Curtobacterium* spp., *Erwinia* spp., *Pantoea* spp., *Pseudomonas* spp., *Sphingomonas* spp., and *Yersina* spp. (Appendix A). These differences were not significant, however, when only untreated grapevines were analyzed (Appendix A), possibly due to a smaller number of replicates. *Raoultella* spp. and *Enterobacter* spp. were the most abundant genera in non-necrotic woody tissues of untreated Alsace grapevines (44% and 39%, respectively, for a total of 83%, Table 4). *Brevundimonas* spp. was the principal genus in non-necrotic tissues of untreated Champagne grapevines (98%, Table 4). Finally, *Curtobacterium* spp. and *Amnibacterium* were the most identified in non-necrotic tissues of untreated Languedoc grapevines (37% and 22%, respectively, for a total of 59%, Table 4). White-rot necrotic woody tissues of untreated Alsace grapevines were almost exclusively colonized by three taxa, *Nocardia* spp., *Brevundimonas* spp., and *Microbacterium* spp. (40%, 18%, 12% for a total of 71 %, Table 4). White-rot necrotic woody tissues of untreated Champagne grapevines were almost principally colonized by three taxa, *Brevundimonas* spp., *Cellulomonas* spp., and *Sphingomonas* spp. (29%, 26%, 16%, for a total of 71 %, Table 4). In Alsace grapevines, border tissues were mostly colonized by *Pantoea* spp., *Rhizobium* spp., and *Curtobacterium* spp. (23%, 27% and 10%, respectively, for a total of 60%, Table 4). *Pseudomonas* spp. and *Raoultella* spp. were more abundant bacterial taxa in untreated Champagne grapevine border tissues (29% and 32%, respectively, for a total of 62%, Table 4). Finally, *Yersinia* spp. and *Soladis* spp. were more abundant in untreated Languedoc grapevine border tissues (40% and 42%, respectively, Table 4).

### 3.5. Effects of Sodium Arsenite on Fungal Microbiota of Grapevine Woody Tissues

Sodium arsenite treatment significantly changed the species composition of fungal microbiota colonizing trunk woody tissues, as shown previously using ANOSIM (Appendix A *p* = 0.002). Differences in relative abundance were plotted for each species/tissue combination by region (Figure 3). A statistical analysis was performed to identify which species significantly differed in abundance between treated and untreated grapevines, by pooling samples from different regions and different tissues (Appendix A). This analysis showed that the fungal species displaying the highest significant reduction in abundance upon sodium arsenite treatment was *F. mediterranea* (Figure 3, Table 3, *p* = 0.001<, Appendix A), irrespective of region and tissue. This fungal species was detected in high abundance in all woody tissues of untreated grapevines from the three regions studied (15–95%, Table 3). Treatment with sodium arsenite significantly decreased its abundance in white-rot necrotic tissues of Alsace and Champagne grapevines (12- and 15-fold, respectively, Table 3), and in border tissues of Alsace, Champagne, and Languedoc grapevines (52-, 35-, and 61-fold, respectively, Table 3). This decrease in abundance was also observed in non-necrotic tissues of Alsace, Champagne, and Languedoc grapevines, although to a lesser extent (2.3-, 25-, and 1.6-fold, respectively, Table 3). Additional statistical analyses were performed for non-necrotic tissues and borders using regions as replicates (Appendix A). These analyses confirmed that the decrease in abundance of *F. mediterranea* in treated non-necrotic tissues and borders was significant (data not shown, *p* = 0.049 in both comparisons). The second fungal species displaying a significant reduction in abundance upon sodium arsenite treatment was *S. vitis* (Figure 3, Table 3, *p* = 0.02, Appendix A). This treatment remarkably decreased its abundance in non-necrotic tissues of Alsace and Languedoc grapevines by 14- and 21-fold, respectively, (Table 3). The third fungal species with a significant decrease in abundance upon sodium arsenite treatment was *Mycena maurella* (Figure 3, Table 3, *p* = 0.02, Appendix A). Treatment with sodium arsenite decreased its abundance to very low levels (0.02%, 1000-fold, Figure 3, Table 3). All the other major changes observed after sodium arsenite treatment corresponded to an increase in fungal species abundance (eight fungal species, with a >80-fold increase in at least one vineyard, Figure 3, Table 3). Only two of these fungi displayed a statistically significant increase in abundance upon sodium arsenite treatment (*Ionotus hispidus*, *Lepiota brunneoincarnata*, Appendix A). *L. brunneoincarnata* was only detected in untreated non-necrotic woody tissues of Languedoc grapevines at a very low abundance (0.3%, Table 3). Treatment with sodium arsenite increased by 72-fold its abundance in non-necrotic tissues (21%, Figure 3, Table 3). *I. hispidus* was only detected in untreated non-necrotic and border woody tissues of Languedoc grapevines at a low abundance (2.6% and 0.2%, respectively, Table 3). Treatment increased by 83-fold its abundance in border tissues (16%, Figure 3, Table 3). Six other fungal species displayed a large increase in abundance. For example, the abundance of *Hypocrea lixii* increased by 330-fold after treatment in non-necrotic tissues of Alsace grapevines (33%, Figure 3, Table 3). Treatment increased by 2000-fold the abundance of *Aspergillus* spp. in white-rot necrotic tissues of Alsace grapevines (67%, Figure 3, Table 3). Treatment increased by 200-fold the abundance of *unknown Agaricales* genus in non-necrotic tissues of Champagne grapevines (61.7%), and by 800-fold the abundance of *Aureobasidium pullulans* in white-rot necrotic tissues of Champagne grapevines (87.3%, Table 3).

### 3.6. Sensitivity to Sodium Arsenite of Fungi Isolated from Grapevine Woody Tissues

Fungal isolates from different species were obtained from grapevine woody tissues in this study (Appendix A). A *F. mediterranea* isolate was sampled from white-rot necrosis borders of an untreated grapevine from Champagne. A *P. chlamydospora* isolate was sampled from a black punctuation of a treated grapevine from Alsace. An *E. lata* isolate was sampled from a black sectorial necrosis of a treated Alsace grapevine. A *Trichoderma* spp. isolate was sampled from black sectorial necrosis of a treated Champagne grapevine. An isolate of *Penicillium* spp. was sampled from white-rot necrosis borders of a treated Champagne grapevine. Another isolate from *Penicillium* spp. was sampled from non-necrotic tissues of a treated Languedoc grapevine. The *F. mediterranea* isolate was the most sensitive of all the fungi tested (Table 5). Other GTD fungi, such as *P. chlamydospora* and *E. lata*, were 15- and 20-fold less sensitive, respectively, than *F. mediterranea* (Table 5). Among the saprophytic fungi, *Trichoderma* spp. was 72-fold less sensitive than *F. mediterranea*. IC50 values for the two *Penicillium* spp. were 347/652-fold less than the *F. mediterranea* IC50 value (Table 5). In addition, these saprophytic fungi continued to grow, even with a high concentration of sodium arsenite (500 mg/L, Table 5). These experiments showed that plant pathogenic fungi were more sensitive to sodium arsenite than saprophytic fungi (40-fold on average, Table 5). In particular, *F. mediterranea* was the most sensitive fungus to sodium arsenite.

## 4. Discussion

This study characterized variations in the taxonomical composition of fungal and bacterial communities colonizing woody tissues of grapevines from the three independent French vineyards expressing esca disease, treated or not with sodium arsenite (Alsace, cultivar Gewürztraminer; Champagne, cultivar Chardonnay; Languedoc, cultivar Merlot). The taxa compositions of these communities were characterized by using metabarcoding, revealing 718 fungal and 3886 bacterial OTUs. Significant differences in fungal microbiota were observed between treated and untreated grapevines, independently of their region or tissue origin. Sodium arsenite, however, had no significant impact on bacterial microbiota.

### 4.1. Fungal Microbiota Colonizing Woody Tissues from Esca-Diseased Grapevines

The fungal microbiota colonizing trunks of untreated grapevines expressing esca were compared according to the type of woody tissue (non-necrotic, white-rot necrosis, black sectorial necrosis, and border) and vineyard (Alsace, Champagne, Languedoc). Analysis showed that *F. mediterranea*, a GTD fungus, was the most abundant species colonizing all trunk woody tissues of diseased grapevines from two of the three vineyards studied (Alsace, Champagne, Table 3). This unexpected dominance of *F. mediterranea* in woody tissues from diseased grapevines, is not limited to white-rot tissues, and this has already been reported [28,41]. A recent study in Portugal (Instituto Superior de Agronomia vineyard in Lisbon) on Cabernet Sauvignon grapevines expressing or not esca-foliar symptoms, showed that the relative abundance of fungi in woody tissues differs according to the organ sampled (trunk, cordon), but not the disease status [41]. Their study also highlighted that *F. mediterranea* was the only species whose relative abundance increased in woody tissues of trunks and arms/cordons of diseased plants (5-fold) compared to healthy ones, reaching up to 35% of all woody tissues. It should be noted that as this study used drilling for wood sampling, it was not possible to split different types of tissue. Another recent study of young French Cabernet Sauvignon grapevines (Bordeaux vineyard) showed that esca was always associated with the presence of white-rot necrotic tissues in cordons [28]. These white-rot necrotic tissues were mainly colonized by *F. mediterranea* (>90%), as observed in the Alsace and Champagne grapevine trunks (Table 3). Our results strongly support these early findings in two unrelated vineyards (Portugal; Bordeaux, France) suggesting that *F. mediterranea* is the main fungal species colonizing woody tissues of esca diseased grapevines. A discrepancy in this trend was observed in Languedoc diseased grapevines whose woody tissues were mainly colonized by *Seimatosporium vitis* (38-55%, Table 3). Still, in these grapevines, *F. mediterranea* was the second most abundant fungus reaching up to 14% in non-necrotic tissues, and 75% in borders between non-necrotic and white-rot necrotic tissues (WR was not sampled and studied in Languedoc grapevines, Table 3). *S. vitis* was also detected in high abundance specifically in non-necrotic woody tissues from Alsace grapevines (23%, Table 3). This fungal species has been reported as a saprophyte or a pathogen of different tree species, as well as grapevine [41,42,43]. In grapevines, it was isolated from necrotic sectors of woody tissues (cankers) of trunks from diseased plants (GTDs) in Hungary, USA California, and Italy [44,45,46,47]. It was also isolated from dead branches of grapevines in Italy [42] and Iran [48]. *S. vitis* is pathogenic to grapevine, as it induces an external local necrosis when inoculated to wounded stems of potted grapevines [44,45,47]. This fungus could be involved in the formation of internal necrosis observed in trunks. Its detection in woody tissues of grapevines with GTDs is quite recent (reports only since 2015), and is variable according to regions, as it was not detected in trunks of grapevines from Champagne (this study) or Bordeaux [28]. However, this fungal species merits further studies to better assess its presence in vineyards and its role in GTDs.

Among other GTD fungi, only *P. chlamydospora* was detected in high abundance (>15%), specifically in borders between white-rot necrotic tissues and non-necrotic tissues in all three vineyards. In these borders, *P. chlamydospora* was the only other fungal species associated with *F. mediterranea.* No other GTD fungi exceeded 5% abundance in any woody tissues, except for *D. seriata*. This fungus, *D. seriata*, was detected in non-necrotic tissues of Alsace and Languedoc grapevines at 5.2% and 7%, respectively, and in sectorial necrosis of Languedoc grapevines up to 18% (Table 3).

### 4.2. Bacterial Microbiota Colonizing Woody Tissues from Esca-Diseased Grapevines

Our experiments highlighted that sodium arsenite has no effect on bacterial communities. Still, significant differences in bacterial taxa composition were observed between vineyards. Some bacteria used as biocontrol agent such as *Pantoea* spp., were identified in non-necrotic tissues of both Alsace and Languedoc grapevines at relatively high abundances (5.6% and 13.7%, respectively, Table 4). Bruez et al. also isolated *Pantoea* from non-necrotic and necrotic woody tissue of trunks from esca-diseased grapevines [49]. Other bacterial taxa identified in our study were Proteobacteria from *Pseudomonas* spp. and *Sphingomonas* spp. genera (Table 4). In grapevine, these genera were determined with high abundance in bunches, leaves, and flowers, but in low abundance in woody tissues [28,50]. In the present study, these genera were observed in low abundance, meaning that these bacteria could be tissue dependent. In young Cabernet-Sauvignon cultivar, Bruez et al. determined *Sphingomonas* sp., and *Pantoea* sp. in the non-necrotic tissue of esca-diseased grapevines. This was based on our results and those from Bruez et al. on Cabernet-Sauvignon cultivar in a Bordeaux vineyard, we hypothesized that the taxonomic composition of bacterial communities from grapevine woody tissues strongly differs according to cultivar or region in France [11,28]. Further samplings are needed to confirm if differences in bacterial taxa composition of trunk woody tissues occur according to the vineyard in different European regions.

### 4.3. Sodium Arsenite’s Impacts on Fungal Microbiota Colonizing Esca-Diseased Grapevine Woody Tissues

In our experiments, all the grapevines previously identified as diseased and treated with sodium arsenite were eradicated, as they did not express any esca-foliar symptoms, whereas all untreated grapevines were diseased, as they expressed esca-foliar symptoms (Table 1). Therefore, the identification of the major changes induced by sodium arsenite on fungal and bacterial microbiota from grapevine woody tissues could shed some light on its effects. Sodium arsenite treatment induced major changes in fungal microbiota, whereas bacterial microbiota stayed unchanged. The first consequence of treatment was a significant decrease in the abundance of three fungal species, namely *F. mediterranea*, *S. vitis*, and *M. maurella* (Table 3). This result suggests that it inhibited the growth and/or affected the survival of these fungal species in grapevine woody tissues. *F. mediterranea* was the most affected fungal species, in particular in white-rot necrotic tissues and borders, as the abundance of this fungal species decreased by 12- to 61-fold in all vineyards (Table 3). The effect of sodium arsenite on *S. vitis* and *M. maurella* was only observed in certain tissues and vineyards, possibly because of the limited distribution of these fungi across tissues/vineyards. The second consequence of the treatment was an increase in abundance of several fungi. However, these increases, although 72-2000-fold, were mostly reported for a single species in a specific tissue and vineyard (Table 3). For example, *L. brunneoincarnata* increased its abundance by 72-fold only in non-necrotic tissues of Languedoc grapevines, reaching a value of 21.6% (the most abundant species in this tissue, Table 3). *H. lixii* increased its abundance by 330-fold only in non-necrotic tissues of Alsace grapevines, reaching a value of 32.9 % (the most abundant species in this tissue, Table 3). A fungal species from an unknown *Agaricales* genus increased its abundance by 800-fold only in the non-necrotic tissues of Champagne grapevines, reaching a value of 61.7% (the most abundant species in this tissue, Table 3). These findings indicate that these increases occurred at random in different vineyards. The only exception to this trend is the increase in abundance of *P. chlamydospora* occurring in borders of grapevines from all vineyards upon sodium arsenite treatment (from 2- to 3.5-fold, Table 3). *P. chlamydospora* seemed to take advantage of the reduction in *F. mediterranea* abundance in borders following sodium arsenite treatment, to colonize massively these tissues. This hypothesis is possibly the best explanation of the changes following sodium arsenite treatment. This supports the following hypothesis as to the mode of action of sodium arsenite on fungal microbiota: sodium arsenite migrates to grapevine woody tissues including necrotic tissues such as white-rot, where it selectively inhibits the growth of fungi such as *F. mediterranea*, but without affecting the growth of other fungi such as *Aspergillus* spp. or *Hypocrea* spp. Such selective inhibition allows other fungi to colonize tissues previously occupied by *F. mediterranea*. In support of this hypothesis, a quantification of sodium arsenite in white-rot necrotic tissues of treated plants revealed a higher level in white-rot necrotic tissues (258 mg As/kg DW) than in other tissues (51 mg As/kg DW in border tissues, 0.2 mg As/kg DW in non-necrotic tissues) [51]. These results showed that sodium arsenite accumulated massively in white-rot necrotic tissues where *F. mediterranea* was mostly located. We also tested whether in vitro sensitivity to sodium arsenite differed among fungi. *F. mediterranea* was the most sensitive species among those tested (Table 5). Saprophytic fungi from woody tissues were 72- to 652-fold less sensitive to sodium arsenite than *F. mediterranea.* These observations were similar to those obtained by Larignon, who reported that *F. mediterranea* is more sensitive to sodium arsenite than other fungi, including *P. chlamydospora* [25]. Altogether, the accumulation of sodium arsenite in white-rot necrotic tissues and the higher sensitivity of *F. mediterranea* to sodium arsenite, strongly suggest that sodium arsenite selectively inhibits *F. mediterranea* located in white-rot tissues, and this massive decrease in *F. mediterranea* eliminated inner necroses and typical esca-foliar symptoms in grapevines.

## 5. Conclusions

Sodium arsenite was efficiently used to protect and cure grapevines from GTDs before its ban. The present metabarcoding study showed sodium arsenite induced major changes in the fungal microbiota colonizing grapevine woody tissues. Its major effect was to decrease massively the abundance of *F. mediterranea* involved in GTDs, in all the tissues and vineyards studied, in particular in white-rot necrotic tissues. This significant decrease in *F. mediterranea* abundance is associated with an accumulation of sodium arsenite in woody tissues and a high sensitivity of *F. mediterranea* to this fungicide. The treatment with sodium arsenite of diseased plant decreases *F. mediterranea* and makes the typical esca-foliar symptoms disappear. This fungus is a key of the development of this complex disease, as proposed for young grapevines (Bruez, 2020). Sodium arsenite, however, did not affect the bacterial microbiota, either non-pathogenic or the pathogenic including bacteria that could be involved in wood degradation. Based on our results, it would be interesting to develop products as efficient as sodium arsenite on *F. mediterranea*. One possible way is to design fungicides mimicking sodium arsenite effects on wood microbiota. This goal should include a good delivery of the fungicide to woody tissues including white-rot and a high toxicity toward *F. mediterranea*.

## Figures and Tables

**Figure 1 jof-07-00498-f001:**
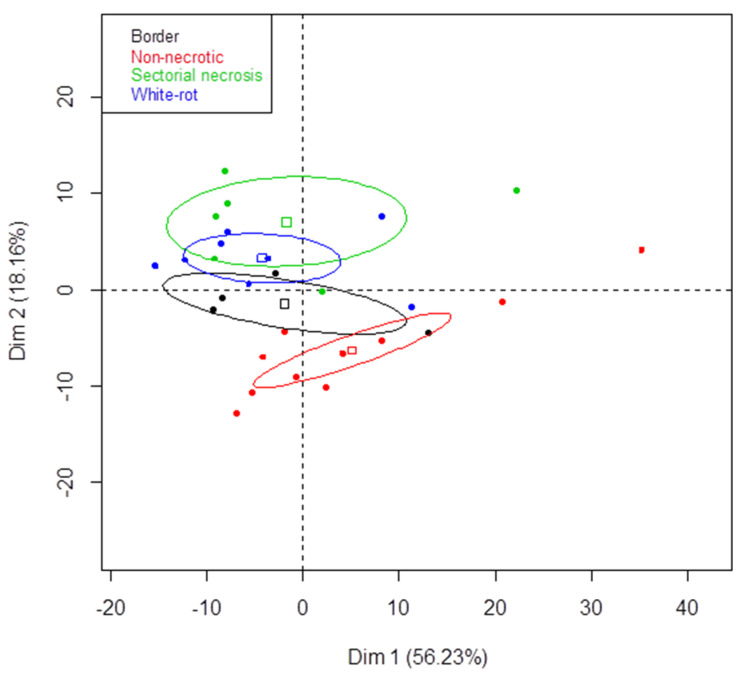
Principal component analysis (PCA) of SSCP profiles of fungal microbiota from woody tissues of untreated grapevines sampled in 2014. SSCP profiles of fungal communities from woody tissues of untreated grapevines from the three regions sampled were analyzed using PCA. The % of total variation explained by each axis is given in brackets. Ellipses represent the 95% confidence intervals for microbiota from each type of tissues (non-necrotic, red; white-rot necrosis, blue; sectorial necrosis, green; border, black).

**Figure 2 jof-07-00498-f002:**
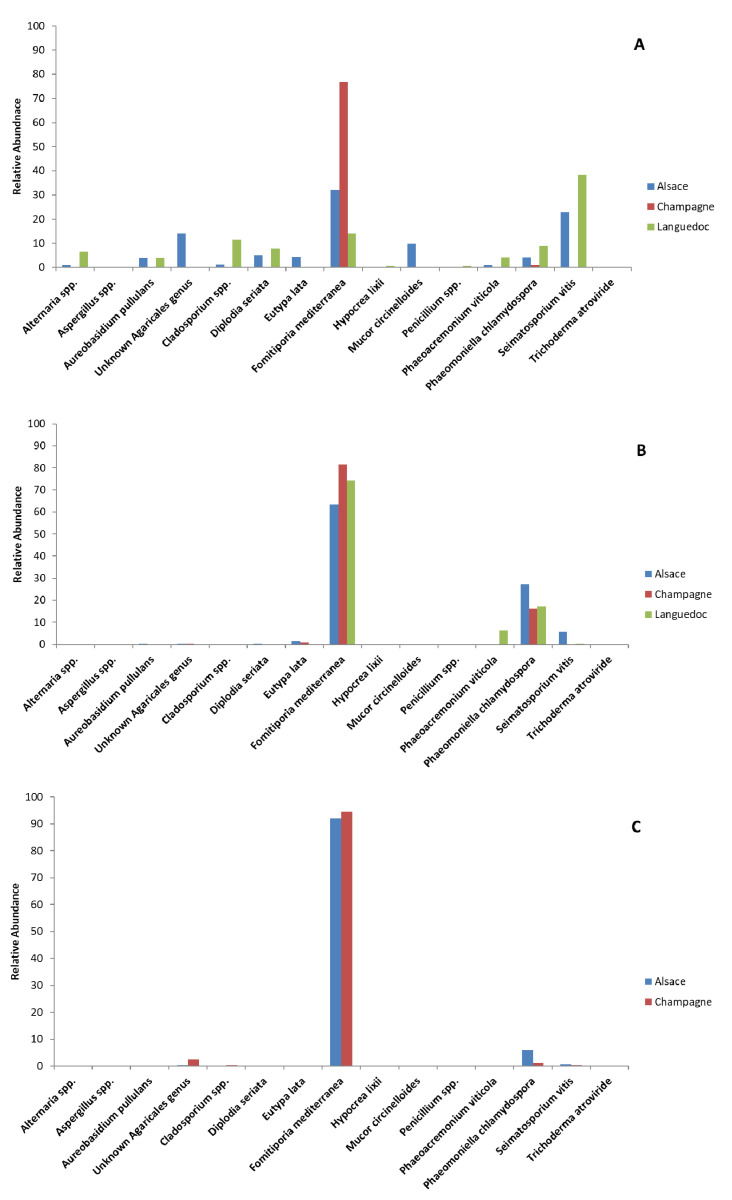
Distribution of fungal taxa in woody tissues of grapevines not treated by sodium arsenite. Data obtained from fungal ITS1 metabarcoding of samples from 2014. %: percentage of OTU reads for a given sample relative to the total number of reads for this OTU. Relative abundance of fungal species in non-necrotic (**A**), border (**B**), and white-rot necrosis (**C**) woody tissues of untreated grapevines from Alsace (blue), Champagne (red) and Languedoc (green).

**Figure 3 jof-07-00498-f003:**
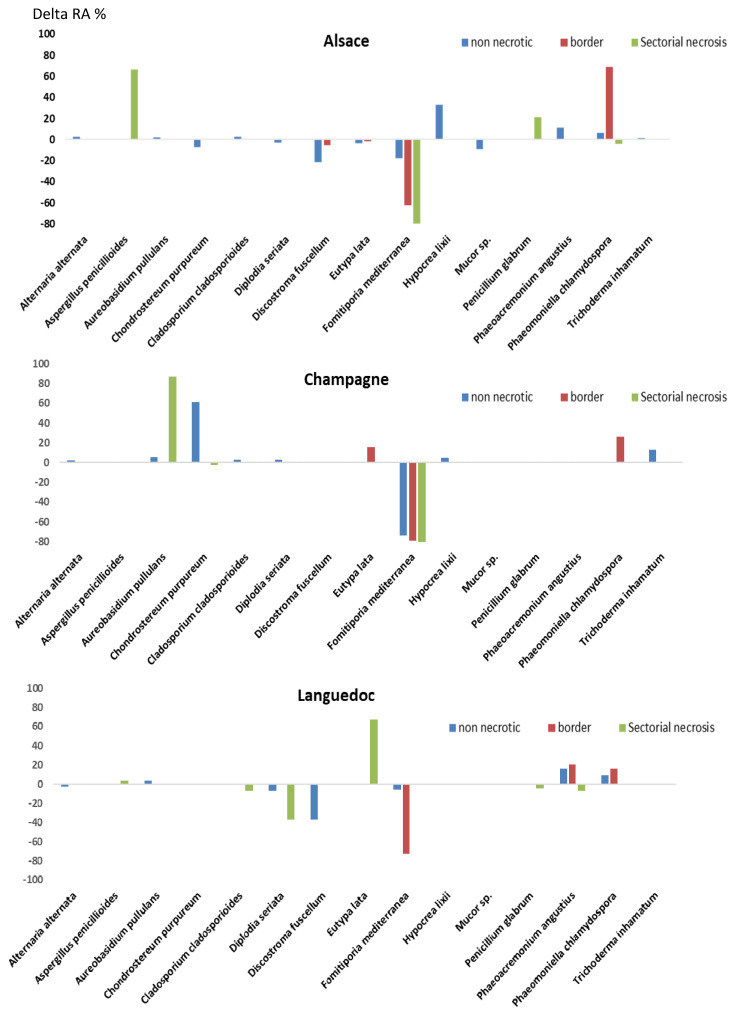
Differences in relative abundance of fungal species between grapevines treated or not by sodium arsenite. Data obtained from fungal ITS1 metabarcoding of samples from 2014. %: percentage of OTU reads for a given sample relative to the total number of reads for this OTU. Differences in relative abundance (%) between treated and untreated grapevines (0: no difference, +100%: increase by 100% after treatment, −100%: decrease by 100% after treatment). Non-necrotic (blue), border (red), and white-rot necrosis (green) woody tissues.

**Table 1 jof-07-00498-t001:** Woody necroses and disease status of grapevines sampled in September 2014. *, % of grapevines displaying each type of necrotic woody tissues **, white-rot necrotic tissue was not in sufficient quantities to enable DNA to be extracted. ***, % of grapevines displaying typical esca-foliar symptoms.

	Alsace		Champagne		Languedoc	
	Treated	Untreated	Treated	Untreated	Treated	Untreated
Non-necrotic (NN)	100% *	100%	100%	100%	100%	100%
Border (NN-WR)	100% *	100%	100%	80%	75%	80%
White-rot necrosis (WR)	100% *	100%	80%	80%	50% **	60% **
Sectorial necrosis (SN)	0% *	100%	0%	0%	100%	100%
Esca-foliar symptoms	0% ***	100%	0%	100%	0%	100%

**Table 2 jof-07-00498-t002:** Taxonomic diversity fungal and bacterial microbiota from woody tissues of grapevines sampled in 2014 and 2015.

Shannon (2014)		Alsace White-Rot	Champagne White-Rot	Languedoc Sectorial Necrosis
Treated	Fungi	1.53	1.17	1.80
	Bacteria	1.39	1.34	3.27
Untreated	Fungi	0.57	0.99	1.60
	Bacteria	4.40	2.37	4.77
Shannon (2015)		Alsace White-rot	Champagne White-rot	Languedoc Sectorial necrosis
Treated	Fungi	1.79	1.51	2.03
	Bacteria	1.59	1.97	4.48
Untreated	Fungi	0.66	0.44	2.21
	Bacteria	2.33	2.15	5.51

**Table 3 jof-07-00498-t003:** The 20 most abundant fungal species identified in woody tissue of grapevines from Alsace, Champagne, and Languedoc, using metabarcoding. Highest percentages are highlighted in gray. % of reads for a single species, in gray values over 10%.

	Treated			Untreated			Treated			Untreated			Treated			Untreated		
	Non-Necrotic	Border	White-Rot	Non-Necrotic	Border	White-Rot	Non-Necrotic	Border	White-Rot	Non-Necrotic	Border	White-Rot	Non-Necrotic	Border	Sectorial Necrosis	Non-Necrotic	Border	Sectorial Necrosis
*Alternaria* spp.	3.8	0.2	0.1	0.9	0.1	0.0	2.1	0.0	0.1	0.0	0.0	0.0	3.7	0.2	0.1	6.4	0.1	0.0
*Aspergillus* spp.	0.2	0.0	66.8	0.0	0.0	0.0	0.0	0.0	0.0	0.0	0.0	0.0	0.0	0.0	0.0	0.0	0.0	0.0
*Aureobasidium pullulans*	6.2	0.1	0.1	3.9	0.5	0.1	5.8	0.1	87.3	0.1	0.1	0.1	7.1	0.1	0.1	4.0	0.1	0.0
Unknown *Agaricales* genus	7.1	1.1	0.4	14.0	0.3	0.3	61.7	0.7	0.2	0.3	0.3	2.5	0.3	0.1	0.2	0.3	0.3	0.2
*Cladosporium* spp.	4.0	0.3	0.1	1.2	0.1	0.1	2.9	0.1	0.3	0.1	0.1	0.4	11.0	0.3	0.2	11.4	0.1	0.0
*Diplodia seriata*	2.3	0.1	0.5	5.2	0.3	0.2	3.1	0.1	0.8	0.2	0.1	0.2	1.1	0.1	14.0	7.8	0.2	18.0
*Eutypa lata*	1.0	0.1	0.4	4.4	1.5	0.2	0.1	16.5	0.8	0.1	0.8	0.1	0.1	0.1	0.1	0.2	0.1	0.1
*Fomitiporia mediterranea*	14.7	1.2	7.2	32.2	63.5	92.1	3.1	2.3	6.2	76.9	81.5	94.5	8.7	1.2	39.6	14.1	74.2	16.8
*Hypocrea lixii*	32.9	0.1	0.2	0.1	0.0	0.0	4.7	0.0	0.1	0.1	0.0	0.1	0.1	0.0	0.0	0.6	0.0	0.0
*Inonotus hispidus*	0.2	0.0	0.0	0.0	0.0	0.0	0.0	0.0	0.0	0.0	0.0	0.0	0.0	16.6	0.0	2.6	0.2	0.0
*Lepiota brunneoincarnata*	0.1	0.0	0.0	0.0	0.0	0.0	0.0	0.0	0.0	0.0	0.0	0.0	21.6	0.0	0.0	0.3	0.0	0.0
*Mucor circinelloides*	0.4	0.0	0.0	9.8	0.0	0.0	0.0	0.0	0.0	0.0	0.0	0.0	0.0	0.0	0.0	0.0	0.0	0.0
*Mycena maurella*	0.2	0.0	0.0	0.0	0.0	0.0	0.0	0.0	0.0	20.4	0.1	0.0	0.0	0.0	0.0	0.0	0.0	0.0
*Penicillium* spp.	0.1	0.0	21.0	0.0	0.0	0.0	0.0	0.0	0.1	0.0	0.0	0.0	0.7	0.0	0.0	0.5	0.0	0.0
*Phaeoacremonium viticola*	12.5	0.4	0.7	1.0	1.0	0.3	1.1	1.0	1.0	0.3	0.2	0.3	20.3	27.3	7.0	4.1	6.5	3.1
*Phaeomoniella chlamydospora*	10.8	96.1	1.5	4.2	42.4	5.8	1.4	42.4	1.6	1.0	16.2	1.1	18.4	33.6	9.6	8.8	17.2	5.7
*Phaeoacremonium fraxinopennsylvanicum*	0.4	0.0	0.0	0.0	36.5	0.1	0.4	36.5	0.0	0.0	0.0	0.1	0.1	0.0	0.0	0.1	0.0	0.0
*Sebacina* spp.	0.3	0.0	0.1	0.0	0.0	0.1	0.0	0.0	0.1	0.0	0.0	0.1	4.9	19.9	0.0	0.0	0.5	0.0
*Seimatosporium vitis*	1.6	0.2	0.8	23.0	0.2	0.6	0.2	0.2	1.3	0.3	0.2	0.3	1.8	0.3	28.9	38.5	0.3	55.8
*Trichoderma atroviride*	1.3	0.1	0.1	0.1	0.1	0.1	13.1	0.1	0.1	0.1	0.1	0.1	0.1	0.1	0.1	0.2	0.1	0.1

**Table 4 jof-07-00498-t004:** The 20 most abundant bacterial taxa identified in woody tissue of grapevines from AlScheme 10. ND: not done.

	Alsace						Champagne						Languedoc					
	Treated			Untreated			Treated			Untreated			Treated			Untreated		
	Non-Necrotic	Border	White-Rot	Non-Necrotic	Border	White-Rot	Non-Necrotic	Border	White-Rot	Non-Necrotic	Border	White-Rot	Non-Necrotic	Border	Sectorial Necrosis	Non-Necrotic	Border	Sectorial Necrosis
*Amnibacterium* sp.	0.0	0.0	0.0	0.0	14.8	6.5	0.0	0.3	0.0	0.0	0.2	0.0	6.5	ND	0.5	22.3	0.0	26.0
*Badyrhizobium* sp.	2.1	1.0	6.4	0.0	0.7	2.9	1.8	0.5	4.9	0.4	11.0	6.0	2.6	ND	0.0	2.6	0.1	0.1
*Brevundimonas* sp.	26.6	0.7	33.5	0.3	9.7	18.3	84.0	5.5	53.8	98.7	4.1	29.1	34.8	ND	1.1	7.3	0.3	1.0
*Buttiauxella* sp.	0.3	1.2	0.0	1.2	0.4	0.1	0.0	0.5	0.0	0.0	0.3	0.0	0.8	ND	0.3	0.4	0.0	0.1
*Cellulomonas* sp.	0.0	0.0	0.0	0.0	0.1	0.3	0.0	0.6	0.0	0.0	0.5	26.4	0.9	ND	0.1	0.5	0.1	0.1
*Chryseobacterium* sp.	0.1	13.5	0.0	0.0	0.3	6.4	4.2	0.4	0.0	0.2	0.2	3.1	1.8	ND	0.0	0.2	5.9	0.1
*Cloacibacterium* sp.	0.0	5.9	0.0	0.0	0.3	6.4	3.7	29.4	0.0	0.1	0.2	4.0	3.0	ND	0.0	0.2	5.8	0.1
*Curtobacterium* sp.	0.3	0.1	0.1	0.1	10.1	1.4	0.1	0.8	0.1	0.1	0.7	0.1	16.9	ND	28.7	37.6	0.1	35.7
*Enterobacter* sp.	0.1	0.2	0.0	39.2	0.2	0.1	0.0	0.5	0.0	0.0	1.6	0.0	0.4	ND	0.1	0.4	0.1	0.1
*Erwinia* sp.	2.4	3.9	0.0	0.4	1.8	0.1	0.1	2.2	0.0	0.0	2.2	0.0	0.9	ND	39.0	1.2	0.2	11.1
*Methylobacterium* sp.	0.5	0.1	3.8	0.0	0.6	0.7	0.1	0.6	7.8	0.0	0.2	4.4	0.7	ND	0.1	0.8	0.0	0.2
*Microbacterium* sp.	7.3	0.2	0.2	0.0	0.2	12.6	0.1	0.2	0.0	0.0	0.7	0.3	5.4	ND	0.2	3.2	0.1	0.1
*Nocardia* sp.	0.9	0.7	0.0	0.0	1.5	40.6	4.5	21.3	0.2	0.0	0.6	3.6	1.4	ND	0.0	1.6	0.0	0.1
*Pantoea* sp.	45.1	68.8	0.1	13.7	23.7	0.2	0.1	2.5	0.1	0.1	1.2	0.1	3.9	ND	21.9	5.6	0.2	6.9
*Pseudomonas* sp.	0.1	1.3	0.1	0.2	5.8	0.2	0.7	30.8	0.0	0.0	29.7	0.0	7.6	ND	5.0	9.1	0.1	2.6
*Raoultella* sp.	1.3	1.3	0.1	44.4	1.5	0.2	0.1	1.0	0.1	0.1	32.4	0.1	1.0	ND	1.3	1.7	0.9	0.5
*Rhizobium* sp.	1.6	0.8	4.9	0.1	27.0	1.2	0.2	0.7	2.2	0.1	12.7	6.9	9.0	ND	1.4	1.4	0.2	14.4
*Sodalis* sp.	0.0	0.1	0.0	0.0	0.2	0.0	0.0	0.3	0.0	0.0	0.2	0.0	0.3	ND	0.0	0.5	45.8	0.1
*Sphingomonas* sp.	11.2	0.2	50.7	0.1	0.9	1.7	0.1	1.7	30.7	0.1	1.2	15.8	1.7	ND	0.2	2.2	0.1	0.4
*Yersinia* sp.	0.0	0.1	0.0	0.0	0.1	0.0	0.0	0.2	0.0	0.0	0.2	0.0	0.5	ND	0.0	0.8	40.1	0.1

**Table 5 jof-07-00498-t005:** Sensitivity of fungal species from grapevine woody tissues to sodium arsenite. Fungal strains were isolated from grapevine woody tissues (see Appendix A) and cultivated on malt agar supplemented with different concentration of sodium arsenite. * Concentration (mg/L) of sodium arsenite required to inhibit fungal mycelial growth at 50% (IC 50) or 90% (IC 90). ** Relative IC50 Fm: sensitivity of fungi tested compared to *F. mediterranea* (IC50). *** Concentration (mg/L) of sodium arsenite required to kill fungal strains. Concentration required to kill the fungus is >500 mg/L. ND: not done.

Strains	IC 50	Relative IC50 Fm	IC 90	LD 100
*Fomitiporia mediterranea*	0.4 *	1 **	7 *	50 ***
*Phaeomoniella chlamydospora*	6	15	53	100
*Eutypa lata*	8	20	46	500
*Trichoderma* sp.	29	72	210	ND
*Penicillium* sp. strain 1	139	347	18	ND
*Penicillium* sp. strain 2	261	652	442	ND

## Data Availability

https://www.ncbi.nlm.nih.gov/sra/PRJNA732838.

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
