# Peer review of "Impacts of Sodium Arsenite on Wood Microbiota of Esca-Diseased Grapevines"

_jof, 2021, doi:10.3390/jof7070498_

Round 1
Reviewer 1 Report
Dear authors.
All minor corrections, suggestions and recommendations are given in the article.

Author Response
Dear reviewer,
the corrections were made adn please find attached the new manuscript.
Abstract
l.21: "analyzed" was changed by "identified"
l.23: "effects" was changed by "impacts"
l. 26: "strongly" was changed by "significantly"
Introduction
The part from line 70 to line 77 was removed from the text.
l.92: "burst" was changed by "bursting"
The resolution of the table 1 was improved.
l.110 and l.112: "et al." was put in italic.
The variety names were written as suggested by the reviewer.
The table 2 was simplified and the big table 2 was added in supplementary data as table S3.
l.238: information was added in the text.
l.268: significantly was removed from the text.
The title of the figure 2 and the name of Y axis was replaced.
The title of the Table 3 was changed.
The title of the Table 4 was changed.
The title of the figure 3 was changed as the reviewer suggested.
l.395: F.mediterranea was put in italic.
l.471: checked
l.491: the dot was added.
l.587: Persic was changed.

Reviewer 2 Report
The manuscript is well designed and it brings an important news to the scientific and expert community in the field of grapevine industry.
There are two points that have to be corrected:
line 121-122 the link on SUB8307285 should be deleted, SUB..... files should not be published in the final manuscript, moreover it is not available through the pasted link
Fig. 3 rework the graphs to be "clean and clear", it is not possible to publish the text crossing over the graphs
Author Response
Dear Reviewer,
the link https://www.ncbi.nlm.nih.gov/sra/PRJNA732838 is working.
The figure 3 was changed.

This manuscript is a resubmission of an earlier submission. The following is a list of the peer review reports and author responses from that submission.
Round 1
Reviewer 1 Report
Dear Authors,
all comments, suggested corrections and recommendations are given in the reviewed manuscript.

Author Response
Check list
Comment 1: l. 2 Impacts?
Answer: the word was changed in the title and the text.
Comment 2: l. 32 usually avoid the terms already presented in the title
Answer: the keywords were changed.
Comment 3: l.47 add the reference
Answer: the references Rusjan et al., 2017 was added.
Comment 4: l. 67-75 usually the description of the experiment is not included… of the study.
Answer: the last paragraph was changed and modified.
Comment 5: l. 80 according to the international nomenclature… quotation marks
Answer: “Gewürztraminer” was modified in all the text.
Comment 6: l. 181, l. 217 titles or subtitles have to be short ; shorten
Answer: the titles have been changed.
Comment 7: l. 186 infected,… symptomatic?
Answer: it was changed by “from esca-diseased grapevines”.
Comment 8: l. 194 among
Answer: « between » was changed by « among »
Comment 9: l. 214-216 the statement with…. The chapter Introduction
Answer: the sentences were put in the Introduction part
Comment 10: l.254 Dot
Answer: the dot was added.
Comment 11: l.259 the
Answer: « the » was added « to the region »
Comment 12: l.289, l. 301, l.302, l. 311 significant already means statistically
Answer: « statistically »was removed from the text.
Comment 13: l. 293, l. 306 significantly, remarkably…
Answer: « dramatically » was changed by « significantly » and « remarkably »
Comment 14: l.331
Answer: « isolate » was removed from the sentence.
Comment 15: l.351
Answer: « by » was added « characterized by using »
Comment 16: l.385 trees species,…
Answer: « species » was added.
Comment 17: l411
Answer: the dot was added.
Comment 18: l.419-421
Answer : the sentence was rewritten
Comment 19 : l.439-447
Answer: Larignon et al. (2018) Showed also that some fungi are sensitive against the sodium arsenite.
Comment 19: l. 467
Answer: « et al. » was removed.
Comment 20: l. 479 significant
Answer: « significant »replaced « strong »
Reviewer 2 Report
The authors designed an interesting study, there are also an interesting findings and results, however there are also parts that must be reworked according to the comments in the attached pdf

Author Response
Comment 1: cannot be established species level
Answer: Thank you for your comments. We appreciate your wish “to be honest”, and it is in the same spirit that we would like to draw your attention to several recent articles. One of these, published in Frontiers Plant Sciences in July 2019 (Del Frari, G et al., Characterization of the Wood Mycobiome of Vitis vinifera in a Vineyard Affected by Esca. Spatial Distribution of Fungal Communities and Their Putative Relation With Leaf Symptoms. Frontiers Plant Sci. 10: 910), studied the woody tissue mycobiome of grapevines. In their study, the authors described fungal species abundance across samples, using ITS1. A second paper on the microbiome from woody grapevine tissues was published even more recently, in Frontiers in Microbiology (Niem et al., 2020), Diversity Profiling of Grapevine Microbial Endosphere and Antagonistic Potential of Endophytic Pseudomonas Against Grapevine Trunk Diseases. Front Microbiol. 11: 477). The Authors used ITS metabarcoding. This is also the case for the paper of Zarraonaindia et al. (2015, The Soil Microbiome Influences Grapevine-Associated Microbiota. Mbio 6: e02527-14.). The last year, Bruez et al. (2020), published an article in a leading journal, Environmental Microbiology describing the fungal species from wood microbiota using also ITS1 for fungal species identification. In all the articles cited above, the authors used ITS metabarcoding to identify fungi at the species level, and 16S metabarcoding to identify bacteria at the genus level, and both types of metabarcoding were applied in the present manuscript. Like all the above mentioned-authors, we performed a BLAST analysis of the ITS sequences retrieved from our metabarcoding data. As a supplementary file, we provide this analysis below for the most abundant fungal species in our study: Fomitiporia mediterranea, Phaeomoniella chlamydopsora and Seimatosporium vitis. This analysis shows that each metabarcoding IT1 sequence had an exact match for a single fungal species, as reflected in each of phylogenetic trees shown. This detailed analysis confirms that our ITS1 sequences duly identified the fungal species concerned.
Comment 2: add a sentence about the influence of bacterial spectra
Answer: the is a sentence already about the bacteria in the abstract : l. 23 : « … while no major effects were observed on bacteria. »
Comment 3: l. 200, « N instead of n »
Answer : Done.
Comment 4: line 413, « dot »
Answer : the dot was added.
Comment 5: l. 475, « delete it, it is not conclusion of this study »
Answer : the sentence was delated.
Comment 6: l.486-487, « based on the results, …on Fomitiporia. »
Answer : the sentence was changed.
Round 2
Reviewer 2 Report
After a thorough repeated reading of the manuscript was found out that the most important points that were pointed out (via previously attached pdf) were not considered at all. The effort was done to increase the quality of the paper however the authors have not been willing to hear the calls to do it.
The main points of the concerns are still the species level determination (if you want to do it, there must be at least parameters about the consensus lenghts - percents of read lenghts used per OTU), for example: similarity >97% supported by the lenght of alignment >95%. And even this cannot be valid for genera as for ex Penicillium, Hypocrea or yeasts. Also many trunk pathogens should be determined using more genes, ITS easily cannot be understand as a marker for species determination. The same for determination of bacteria and 16S (V5-V6) marker.
There are too few details of analyses and their results. The SRA number is still blind.
The final consideration of the acceptation rely on the editors but I would recommend to reject the manuscript.